# Rapid Detection of the Strawberry Foliar Nematode *Aphelenchoides fragariae* Using Recombinase Polymerase Amplification Assay with Lateral Flow Dipsticks

**DOI:** 10.3390/ijms25020844

**Published:** 2024-01-10

**Authors:** Sergei A. Subbotin

**Affiliations:** Plant Pest Diagnostic Centre, California Department of Food and Agriculture, 3294 Meadowview Road, Sacramento, CA 95832-1448, USA; sergei.a.subbotin@gmail.com

**Keywords:** California, diagnostics, fern, ITS rRNA gene, rice, species-specific primer, strawberry

## Abstract

Rapid and reliable diagnostic methods for plant-parasitic nematodes are critical for facilitating the selection of effective control measures. A diagnostic recombinase polymerase amplification (RPA) assay for *Aphelenchoides fragariae* using a TwistAmp^®^ Basic Kit (TwistDx, Cambridge, UK) and AmplifyRP^®^ Acceler8^®^ Discovery Kit (Agdia, Elkhart, IN, USA) combined with lateral flow dipsticks (LF) has been developed. In this study, a LF-RPA assay was designed that targets the ITS rRNA gene of *A. fragariae.* This assay enables the specific detection of *A. fragariae* from crude nematode extracts without a DNA extraction step, and from DNA extracts of plant tissues infected with this nematode species. The LF-RPA assay showed reliable detection within 18–25 min with a sensitivity of 0.03 nematode per reaction tube for crude nematode extracts or 0.3 nematode per reaction tube using plant DNA extracts from 0.1 g of fresh leaves. The LF-RPA assay was developed and validated with a wide range of nematode and plant samples. *Aphelenchoides fragariae* was identified from seed samples in California. The LF-RPA assay has great potential for nematode diagnostics in the laboratory with minimal available equipment.

## 1. Introduction

The strawberry foliar nematode *Aphelenchoides fragariae* is an important pest of strawberry plants; it causes ‘strawberry crimp’ disease [1]. *Aphelenchoides fragariae* was described by Ritzema Bos [2] from specimens extracted from strawberry plants sent to him from England. The neotype of this species proposed and described by Allen [3] came from strawberries in Escalon, CA, USA. The nematode attacks above-ground parts of the plant and causes malformations, including twisting of the shoots and puckered undersized leaves with crinkled edges, reddened petioles and discolored areas that have hard, rough surfaces. The nematode damage symptoms are easily confused with symptoms of powdery mildew or infections with plant pathogenic bacteria. *Aphelenchoides fragariae* can severely impact the yield of strawberries, as heavily infected plants do not grow normally or produce fruit [4,5]. Over 250 plants in 47 families, including ornamental ferns and other ornamental plants, are also considered to be hosts of *A. fragariae* [3,4,5,6,7,8,9,10,11,12,13]. In California, *A. fragariae* has been recorded on 27 plant species [14]. This nematode is widely distributed in the USA, Canada, Japan and Europe and currently reported in 37 countries [7,15]. Interestingly, the strawberry foliar nematode may be endo- or ectoparasitic, but it can also be mycetophagous [16], attributes that enhance its survival in the absence of a host crop and which limit and complicate management options.

The two nematode species most associated with damage in California strawberries are the foliar nematode *A. fragariae* and the northern root-knot nematode *Meloidogyne hapla*. Strawberries are an important crop in the United States with about 90% of their national production occurring in California. Strawberries are California’s third highest grossing crop, valued at USD 3.02 billion in 2021. Strawberry production in California occurs primarily along 500 miles of coastal regions between San Diego in the south to Monterey Bay in the central coastal region [17]. Besides the survival of *A. fragariae* in the mycetophagus state, the western sword-fern *Polystichum munitum*, native to western North America from Alaska to Mexico, is often heavily infected by *A. fragariae* in California coastal forests. The high humidity of coastal areas provides very favorable conditions for a rapid increase in the foliar nematode population and its dispersal from infected natural areas to strawberry plantations.

The strawberry foliar nematode is the subject of a regulatory program in California. Strawberry nursery stock should be inspected and tested for nematodes through the California Strawberry Registration and Certification Program run by the California Department of Food and Agriculture and the Foundation Plant Services of the University of California, Davis [18]. The precise and rapid detection of regulated nematodes is the first and most essential step in the regulatory program.

The morphological identification of *A. fragariae* is complex and requires the microscopic examination of adult nematodes. This species can be distinguished from all other *Aphelenchoides* species by its slender body, lateral field with two incisures and tail terminus with a single mucro [15,19]. Distinguishing between foliar nematodes and fungal-feeding *Aphelenchoides* species based on morphological characteristics is quite problematic. Only molecular methods can provide reliable and rapid diagnostic tools for their differentiation.

Sequences of the nuclear ribosomal genes 18S rRNA, ITS rRNA and D2-D3 of 28S rRNA are clearly differentiate *A. fragariae* from other nematodes [20,21,22,23,24,25,26,27]. Ibrahim et al. [28] were the first to provide PCR-ITS-RFLP profiles for *A. fragariae* and other *Aphelenchoides* spp. McCuiston et al. [21] was the first to develop conventional PCR with species-specific primers designed on the ITS rRNA gene polymorphism for nematode detection in plant materials. Rybarczyk-Mydłowska et al. [22] designed a real-time PCR SYBR Green assay using species-specific primers, based on 18S rRNA gene polymorphism, for the quantitative detection of *A. fragariae* and other *Aphelenchoides* spp. Although these PCR assays are very reliable, new DNA amplification techniques may provide easier and more rapid nematode detection.

Recombinase polymerase amplification (RPA) is a relatively new isothermal in vitro nucleic acid amplification technique. It has been adopted as a novel molecular technology for simple, robust, rapid, reliable and low-resource diagnostics for different organisms [29]. RPA has several advantages over PCR-based methods for plant-parasitic nematode detection: (i) it does not require thermal cycling and can be used in areas with minimal laboratory infrastructure and run by personnel with minimal technical experience; (ii) it is more sensitive than PCR; (iii) sample processing does not require DNA extraction; (iv) amplicons may be detected, at the endpoint or in real-time, within 8 to 30 min. RPA assays have shown high sensitivity and specificity for detecting various agriculturally important plant-parasitic nematodes, including *Meloidogyne hapla* [30], but have not been developed for diagnostics in *A. fragariae*.

In this study, an LF-RPA assay was developed for the detection of *A. fragariae* from plant and nematode DNA samples and for crude nematode extracts. Species-specific primers and a probe were designed based on the polymorphism of the ITS ribosomal RNA gene sequences.

## 2. Results

### 2.1. Nematode Samples

Twenty-seven isolates of *A. fragariae* were used to develop and validate the LF-RPA diagnostic assay. Nematodes of this species were collected from various plants in the USA (California, Connecticut, North Carolina, Washington), Russia, Germany and New Zealand (Figure 1; Table 1). *Aphelenchoides fragariae* was identified in rice seeds from California, and *A. smolae* was found in a strawberry sample from OR, USA.

### 2.2. RPA Primers and Probe Design

Several ITS rRNA gene sequences of *A. fragariae* and other *Aphelenchoides* species were downloaded from GenBank and aligned with ClustalX. Regions with high sequence dissimilarity between *A. fragariae* and other *Aphelenchoides* spp. were assessed, and four species-specific *A. fragariae* candidate primers and one probe were manually designed. The Blastn search of these species-specific candidate primer sequences and probe sequences showed high identity (100% similarity) only with the ITS rRNA fragments of *A. fragariae* deposited in GenBank.

### 2.3. RPA Detection

Four primer combinations were screened for the best performance under the same RPA conditions. The species-specific forward AfragF2-ITS and reverse AfragR1-ITS primers were found to be optimal with a clearly visible band. This primer set reliably and specifically amplified the target gene fragment on a gel, approximately 194 bp in length (Figure 2). The final sequences of the primers and probe used for the assays are listed in Table 2 and are indicated in the ITS rRNA gene alignment in Figure 3. Sometimes, non-specific weak bands of other sizes with this primer set were observed in experiments with other nematode samples or a negative control.

### 2.4. LF-RPA Assay

The workflows of LF-RPA detection assay for the strawberry foliar nematode *Aphelenchoides fragariae* are given in Figure 4.

#### 2.4.1. Specificity Testing

The RPA assay was tested for specificity using DNA extracted from eight populations of *A. fragariae*, six *Aphelenchoides* spp., six species belonging to other genera from the order Aphelenchida and an unidentified anguinid nematode parasitizing fern, *P. munitum* (Table 1). The RPA results showed high specificity to *A. fragariae* only, and no positive reactions were observed against any other nematodes. Positive test lines on the LF strips were observed for all *A. fragariae* samples, whereas the samples with other nematode species showed only a control line (Figure 5A).

#### 2.4.2. Sensitivity Testing

The sensitivity assay evaluated the specimen number detection limit for a crude nematode extract and for DNA extracted from strawberry leaves and nematodes. Two-fold serial dilutions of crude nematode extract were prepared with a range between 0.5 and 0.001 nematode per reaction tube. The reliable detection level of *A. fragariae* was estimated to be 0.03 nematode per reaction tube, although weak test bands were also visible in some replicates at lower dilutions (Figure 5B). DNA extracted from 0.1 g of healthy strawberry leaves with 1, 5, 10, 15 and 20 nematodes was used in other sensitivity studies. RPA assays reliably detected this nematode species with plant DNA extracted from 0.1 g of healthy strawberry leaves containing 5 or more *A. fragariae* specimens (Figure 6A). Thus, reliable sensitivity was estimated at 0.3 nematode per reaction tube using plant DNA extracts. No positive results were obtained from the testing of crude plant extracts without target nematodes.

#### 2.4.3. Testing of Plant Herbarium Materials

*Aphelenchoides fragariae* detection was confirmed using DNA extracts obtained from 20 dried leaf samples previously identified as infected by this nematode. Nematode-infected plants were collected and identified by Dr. D. Sturhan in Germany and New Zealand and preserved as herbarium specimens (Table 1). Positive test lines on the LF strips were observed for all DNA samples extracted from infected plant leaves only, whereas DNA samples without nematodes showed only a control line (Figure 6B).

#### 2.4.4. Testing of Field Samples

Detection of the strawberry foliar nematode was confirmed in all strawberry samples with leaves and stolons in which fern leaves infected with *A. fragariae* were added. All extracts from the strawberry samples using Baermann funnels contained more than 30 specimens of non-target nematodes (cephalobids, rhabditids and tylenchids), and strawberry samples with target nematodes contained more than 5 moving *A. fragariae* specimens, which were visible under a binocular microscope. The samples were used to prepare crude nematode extracts. All samples containing *A. fragariae* showed strong positive test lines on the LF strips in RPA assays, whereas extracts from strawberry samples containing only non-target nematodes gave only a control line on the LF strips (Figure 7). No false negative or false positive reactions were observed in this experiment.

## 3. Discussion

The LF-RPA assay developed and tested in this study is a simple, fast and sensitive method for the detection of the strawberry foliar nematode, *A. fragariae*. The assay allows for the detection of this species from crude nematode extracts, nematode DNA and DNA extracts from infected plant tissues. The LF-RPA assay has some important advantages over our previously developed PCR method of *A. fragariae* detection [21], the first being that it uses crude nematode extract for the analysis instead of DNA extracts, which are required for PCR assays. The second advantage is that results are available for up to 25 min for RPA vs. more than 3.0 h for PCR assay including DNA extraction, PCR and electrophoresis.

The present *A. fragariae* diagnostic assay and our recently published *Meloidogyne hapla* diagnostics assay [30] can be used to detect these plant-parasitic nematodes in support of the strawberry certification program. These LF-RPA nematode diagnostic assays could also be performed without any special equipment.

The LF-RPA assay developed in this study is for detection of *A. fragariae* only. The testing did not reveal any false positive results with other nematode samples, including samples with an unidentified anguinid nematode parasitizing the fern, *P. munitum*. This putative new anguinid species found in Washington state, in rainforests, causes necrotic symptoms on leaves similar to those induced by the strawberry foliar nematode.

Diagnostic specificity of an assay facilitates the detection of a target organism even in the presence of non-target species that are potentially cross-reactive. The selection of appropriate species-specific primer and probe sequences that match the genomic region of the target species is critical for assay design. For example, recent extensive testing of primers and a probe proposed by Wang et al. [31] for an RPA diagnostic assay for *Globodera rostochiensis* with a wider range of *Globodera* and other cyst nematode species showed that the assay is not specific. The results of this testing showed that the proposed ITS rRNA gene putatively specific primers and the probe gave positive reactions not only with *G. rostochiensis*, but also with other *Globodera* and representatives of the genus *Punctodera* parasitising grasses [32]. It has been known that complementarity between primers and templates is often crucial for DNA amplification. It is likely that lower reaction temperatures during RPA might also lead to less specific DNA hybridization, compromise primer specificity, and may have a significant effect on the occurrence of false positive results despite mismatches [32].

In the present assay, the *A. fragariae*-specific primers have many mismatch differences with non-target *Aphelenchoides*, and testing did not reveal any false positive reactions with other non-target nematodes. However, the in silico and laboratory tests did not include two species, *A. blastophthorus* and *A. saprophilus*, that have close phylogenetic relationships with *A. fragariae*. Unfortunately, the ITS rRNA gene sequences of these two species are not currently available in GenBank, and their DNA was also not available for the present study. Further testing with other *Aphelenchoides* species is needed for confirmation of the high specificity of the described LF-RPA assay.

Several *Aphelenchoides* species are known to parasitize or to be associated with strawberry plants: *A. besseyi*, *A. bicaudatus*, *A. blastophthorus*, *A. fragariae*, *A. pseudobesseyi*, *A. pseudogoodeyi*, *A. ritzemabosi* and *A. rutgersi* [33,34,35,36,37,38,39]. In this study, *A. smolae* was reported to be a nematode species associated with strawberry for the first time. This species was recently isolated and described from medium soil and tissues of *Lilium orientalis* bulbs imported in China from the Netherlands [40].

In the results of our study, we also identified *A. fragariae* from fern, *Polystichum munitum* and rice seeds. This is the first molecular identification of this nematode in this fern species in California. Previously, *A. fragariae* on *P. munitum* were found in several western states [41] and California. To the best of our knowledge, it is also the first report of *A. fragariae* from rice. Presently, only *A. oryzae* and *A. pseudogoodeyi* belonging to the *A. besseyi* species complex are known to be parasites of rice [37,39]. In California, nematodes from the *A. besseyi* species complex were detected in a fungal culture of *Sclerotium oryzae*, which caused stem rot of rice in 1963. The fungus was collected from a rice field in Butte County that was used by a research facility that exchanged seeds with areas in the southeastern USA. In the last 20 years, there have been very occasional *Aphelenchoides* detections in Butte, Colusa, Sutter, Yolo and Yuba counties during phytosanitary inspections of rice for export [42]. The finding of *A. fragariae* in rice seeds will alert plant pathologists and nematologists to the necessity for further surveys of this pest in rice fields in California. The *Aphelenchoides fragariae* RPA-LFA assay, together with another novel *A. besseyi* species complex assay, that combines the RPA and CRISPR/Cas12a methods [43] could be used in rapid diagnostics of these *Aphelenchoides* pests in plant and soil samples.

## 4. Materials and Methods

### 4.1. Nematode Samples

*Aphenchoides fragariae* isolates and other nematodes used in the present study were obtained from various sources (Table 1). Juvenile stages and adult stages of the strawberry foliar nematode were extracted from fresh leaf samples of different plants using a standard Baermann funnel method [44]. Extracted nematodes were morphologically and molecularly identified. A herbarium collection of *A. fragariae*-infected plant leaves collected in Germany, New Zealand and other locations during 1960–2000 were obtained from Dr. D. Sturhan (BBA, Münster, Germany), who identified the nematode morphologically. The herbarium materials were kept at room temperature until this study, and then, used for plant DNA extraction. Plant materials were also provided by Dr. J.L. McCuiston and identified as nematode-infected in our previous study [21]. DNA of several other nematodes (*A. besseyi*, *A. oryzae*, *A. pseudobesseyi*, *A. ritzemabosi*, *A. smolae*, *Aphelenchoides* sp., *Aphelenchus* sp., *Bursaphelenchus fraudulentus*, *B. mucronatus*, *B. cocophilus* and *Laimaphelenchus hyrcanus*) were also used in assay specificity experiments (Table 1). These species were also identified by molecular methods [37,38,39].

For assay validation with field samples, twelve healthy strawberry plants were provided to the CDFA Nematology lab by California growers. Approximately 20 g of plant tissues from each strawberry sample were placed in Baermann funnels in mist chambers. One gram of infected fern leaves was added to the funnels of half of these samples (Figure 8). The water gradually filled the collection tubes and overflowed slowly enough that nematodes remained in the bottom of the tube. All samples were incubated under the mist for 2 days, and then, each collection tube was carefully removed from its funnel without disturbing the contents. A large pipette was used to draw off the water carefully from each tube to avoid stirring up nematodes that had settled to the bottom, as described by Ayoub [44]. Extracts from all samples were visually inspected under a binocular microscope to reveal the presence of moving *Aphelenchoides* specimens and non-target nematodes. The samples were used to prepare crude nematode extracts.

### 4.2. Preparation of Nematode Crude Extract and DNA Extrication

Nematode crude extracts, nematode DNA and DNA from plant tissues were used for development and validation of the LF-RPA diagnostic assay.

For nematode crude extracts, live nematode specimens were placed into a drop of distilled water on a glass slide and cut by a stainless-steel dental needle under a stereo microscope. Cut nematodes were transferred in water suspension into a 0.2 mL PCR tube. Extracts from 20 adults or fourth juvenile stage nematode specimens in 40 μL of water were used to make a series of two-fold sequential dilutions to test the sensitivity of the assay.

Nematode DNA was extracted from several specimens. Nematodes were placed in 20 μL ddH_2_O on a glass slide and cut by a stainless-steel dental needle under a stereo microscope. Cut nematodes in water suspension were transferred into a 0.2 mL Eppendorf tube, and then, three μL of proteinase K (600 μg/mL) (Promega, Madison, WI, USA) and 2 μL of 10× PCR buffer (Taq PCR Core Kit, Qiagen, Germantown, MD, USA) were added to each tube. The tubes were incubated at 65 °C (1 h) and 95 °C (15 min) consecutively. After incubation, the tubes were centrifuged and kept at −20 °C until use.

DNA from infected and healthy control plant leaves was extracted using the Qiagen DNeasy Plant Mini Kit following the manufacturer’s protocol. Total genomic DNA was eluted in a final volume of 30 μL elution buffer and stored at −20 °C. Leaf tissues (0.1 g) were used for each extraction. In total, 1, 5, 10, 15 and 20 nematodes were also added into tubes with uninfected strawberry leaves (0.1 g), and then, used for DNA extraction as described above. These preparations were used to test the sensitivity of the assay for the detection of nematodes in plant tissues.

For validation of the assay with field samples, live nematodes in 300 mL of water obtained from each sample were added into individual tubes and homogenized with an ultimate laboratory homogenizer, Mini Bead Beater 1 (BioSpec Products, Bartlesville, OK, USA) and one glass bead (5 mm) for 30–60 s to obtain nematode extract.

### 4.3. Nematode Molecular Identification

The ITS rRNA and D2-D3 expansion segments of the 28S rRNA gene were amplified and sequenced from nematode isolates to confirm their species identity. PCR protocols were used as described by McCuiston et al. [21]. The following primer sets were used for PCR: (i) the forward D2A (5′-ACA AGT ACC GTG AGG GAA AGT TG-3′) and the reverse D3B (5′-TCG GAA GGA ACC AGC TAC TA-3′) primers for amplification of the D2–D3 expansion segments of the 28S rRNA gene [45] and (ii) the forward TW81 (5′-GTT TCC GTA GGT GAA CCT GC-3′) and the reverse AB28 (5′-ATA TGC TTA AGT TCA GCG GGT-3′) primers for amplification of the ITS1-5.8-ITS2 rRNA gene [46]. PCR products were purified using a QIAquick PCR Purification Kit (Qiagen, USA) and directly sequenced with the primers mentioned above or cloned using a pGEM-T Vector System II kit (Promega, Fitchburg, WI, USA), and then, the clones were sequenced. Sequencing was performed by Genewiz Inc. (Berkeley, CA, USA). Obtained sequences were compared with those deposited in GenBank using a Blastn search [47]. New sequences were deposited in the GenBank database under accession numbers OR685296-OR685297 (ITS rRNA gene) and OR691588-OR691594 (28S rRNA gene).

### 4.4. RPA Primer and Probe Design and Testing

Two forward and two reverse RPA primers specific to *A. fragariae* were manually designed based on species sequence polymorphisms in the ITS rRNA gene. Primers were synthesized by Integrated DNA Technologies, Inc. (Redwood City, CA, USA). Primers were screened in different combinations using the TwistAmp^®^ Basic kit (TwistDx, Cambridge, UK). Reactions were prepared according to the manufacturer’s instructions. The lyophilized reaction pellets were suspended in 29.5 μL of rehydration buffer, 2.4 μL each of forward and reverse primers (10 μM) (Table 2), 1 μL of DNA template or nematode extract and 12.2 μL of distilled water. For each sample, 2.5 μL of 280 mM magnesium acetate was added to the lid of the tube and the lids were closed carefully. The tubes were inverted 10–15 times and briefly centrifuged to initiate reactions simultaneously. Tubes were incubated at 39 °C (4 min) in a MyBlock Mini Dry Bath (Benchmark Scientific, Sayreville, NJ, USA), and then, they were inverted 10–15 times, briefly centrifuged and returned to the incubator block (39 °C) for 20 min. Sample tubes were then placed in a freezer to stop the reaction. Amplification products were purified with a QIAquick PCR Purification Kit (Qiagen, Germantown, MD, USA). Five microliters of purified product were run in a 1% TAE buffered agarose gel (100 V, 60 min) and visualized with a Gel Green stain. The primer set was selected based on amplification performance. The probe was designed based on species sequence polymorphisms in the ITS rRNA gene. RPA primers and probe were synthetized at Biosearch Technologies (Novato, CA, USA).

### 4.5. LF-RPA Assay

The LF-RPA assay was carried out using an AmplifyRP^®^ Acceler8^®^ Discovery Kit (Agdia, Elkhart, IN, USA). The reaction mixture for each RPA assay was prepared according to the manufacturer’s instructions: The lyophilized reaction pellet was suspended with a mixture containing 6 µL of the rehydration buffer, 2 µL of distilled water, 0.45 µL each of forward and reverse primers (10 µM), 0.15 µL of the probe (10 µM) and 0.5 µL of magnesium acetate. One microliter of the DNA template or extract was added to a reaction tube. The reaction tubes were incubated at 39 °C in a MyBlock Mini Dry Bath (Benchmark Scientific, Edison, NJ, USA) for 20 min. For visual analysis with Milenia^®^ Genline Hybridetect-1 strips (Milenia Biotec GmbH, Giessen, Germany), 120 µL of HybriDetect assay buffer was added to a reaction tube, and then, a dipstick was placed in this mixture. Visual results were observed within 3–5 min, and then, photographed. The amplification product was indicated by the development of an intensively colored test line (lower) and/or a separate control line (upper) to confirm that the system worked properly. Three replicates of each variant were performed for sensitivity and specificity experiments.

## Figures and Tables

**Figure 1 ijms-25-00844-f001:**
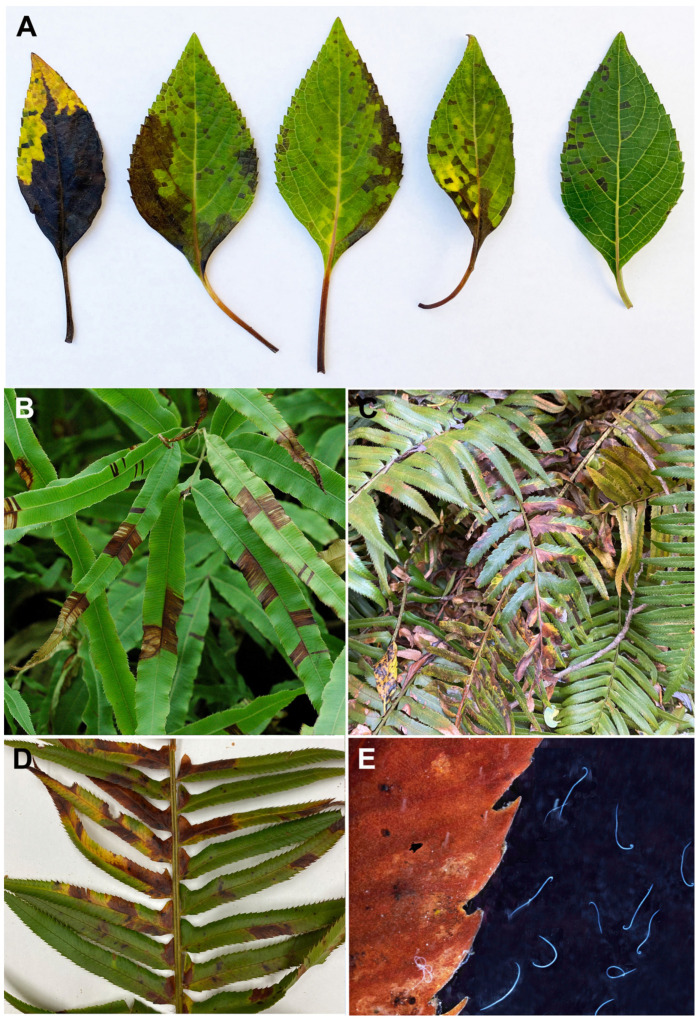
Symptoms of *Aphelenchoides fragariae* infection. (**A**) leaves of *Salvia* sp.; (**B**) *Pteris* plants; (**C**) *Polystichum munitum* plants in Mendocino County, California; (**D**) leaves of *P. munitum*; (**E**) nematodes extracted from *P. munitum* leaves.

**Figure 2 ijms-25-00844-f002:**
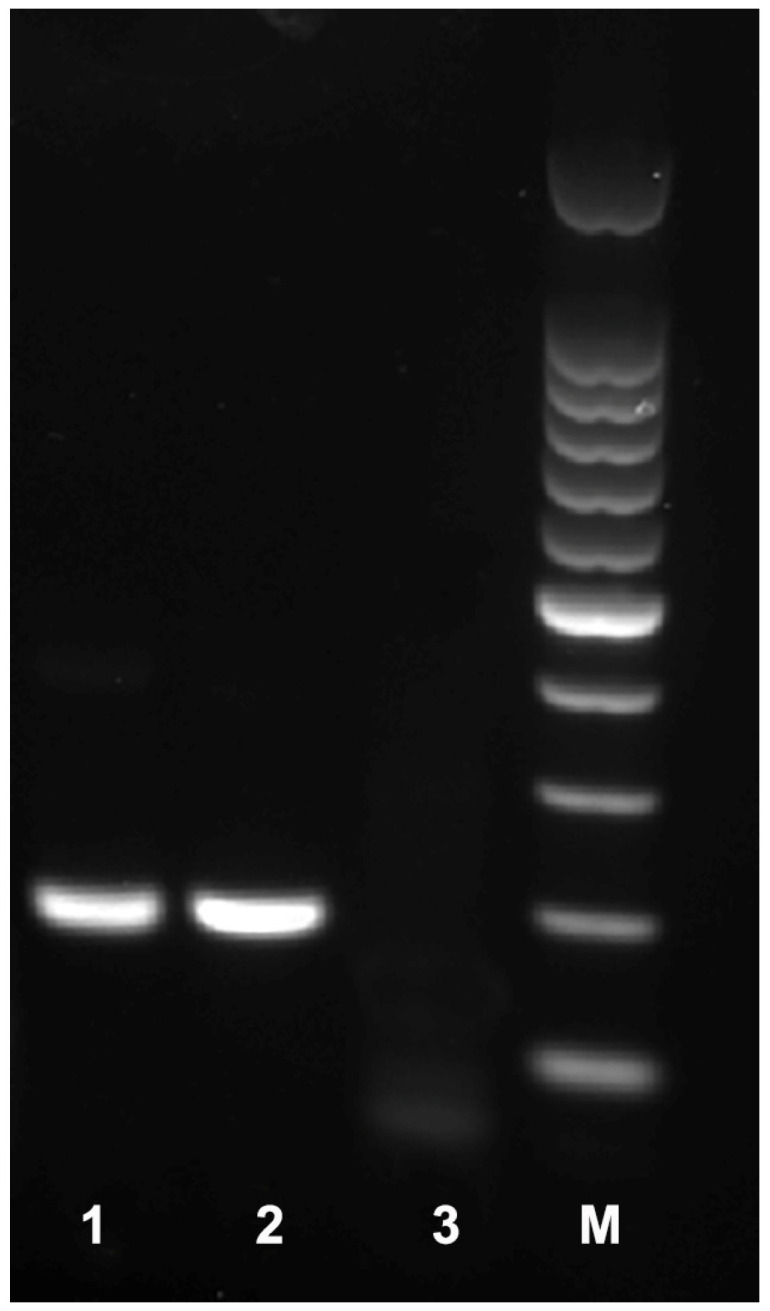
RPA products on agarose gel amplified with AfragF2-ITS and AfragR1-ITS primers using TwistAmp^®^ Basic kit. Lanes: 1: *Aphelenchoides fragariae* (sample CA38); 2: *A. fragariae* (CA53); 3: negative control (no DNA); M: 100 bp DNA marker (Promega, Madison, WI, USA).

**Figure 3 ijms-25-00844-f003:**
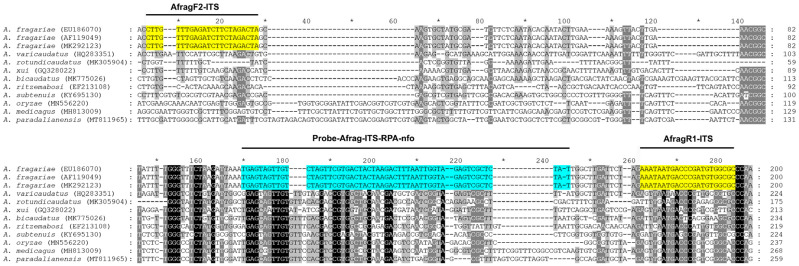
The fragment of the ITS rRNA gene sequence alignment for *A. fragariae* and several *Aphelenchoides* species with positions of RPA primers (yellow) and probe (blue) used in the present assay.

**Figure 4 ijms-25-00844-f004:**
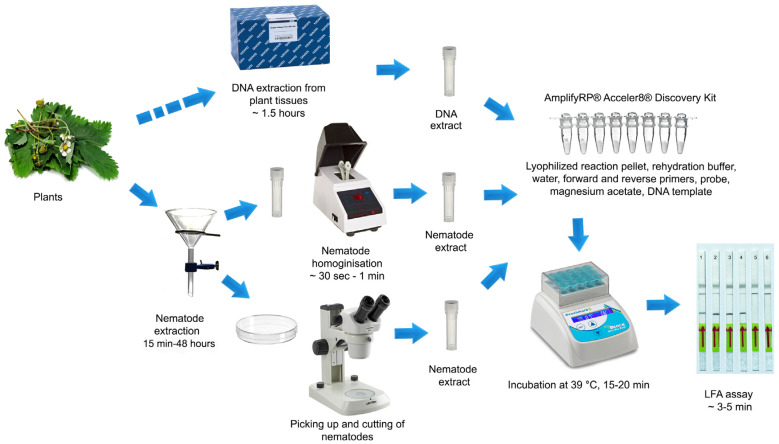
Workflows of LF-RPA detection assay for the strawberry foliar nematode *Aphelenchoides fragariae* used in this study.

**Figure 5 ijms-25-00844-f005:**
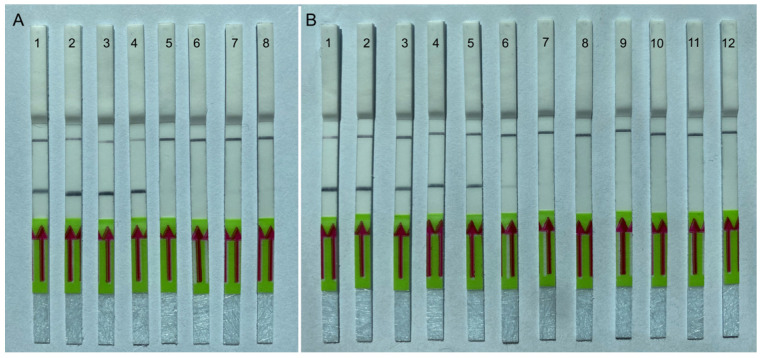
Lateral flow recombinase polymerase amplification (LF-RPA) assay with examples of lateral flow strips. (**A**) Specificity assay with DNA samples of different nematodes. Strips: 1–4: *Aphelenchoides fragariae* (CA30, CA53, CD3764, CD3774); 5: *Bursaphelenchus fraudulentus* (CD2935); 6: *Laimaphelenchus hyrcanus* (CD3646); 7: *A. pseudobesseyi* (CD3702); 8: negative control (no DNA). (**B**) Sensitivity assay with crude nematode extract of *A. fragariae* (CD754). Strips: 1: 0.5 nematode per tube; 2: 0.25 nematode per tube; 3: 0.125 nematode per tube; 4: 0.06 nematode per tube; 5: 0.03 nematode per tube; 6: 0.015 nematode per tube; 7: 0.007 nematode per tube; 8: 0.004 per nematode tube; 9: 0.002 nematode per tube; 10: 0.001 nematode per tube; 11, 12: negative control (no DNA).

**Figure 6 ijms-25-00844-f006:**
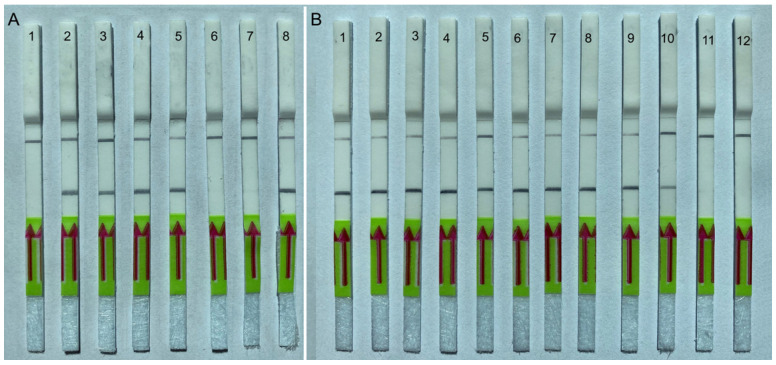
Lateral flow recombinase polymerase amplification (LF-RPA) assay with examples of lateral flow strips. (**A**) Sensitivity assay with DNA extracted from samples containing strawberry leaves and different numbers of *A. fragariae* (CD3787) specimens. Strips: 1: 0.1 g strawberry leave + 1 nematode; 2: 0.1 g strawberry leaves + 5 nematodes; 3: 0.1 g strawberry leaves + 10 nematodes; 4: 0.1 g strawberry leaves + 15 nematodes; 5: 0.1 g strawberry leaves + 20 nematodes; 6: 0.1 g strawberry leaves; 7: negative control (no DNA); 8: positive control (sample CD3787). (**B**) Diagnostic assay with DNA extracted from infected plant leaves. Strips: 1: *Polypodium californica* (CD3794); 2: *Lantana* sp. (CD3795); 3: *Pteris* sp. (CD3796); 4: *Anemone hupehensis* (CD3797); 5: *Heuchera* sp. (CD3798); 6: *Anemone x hybrida* (CD3799); 7: *Mimulus moschatus* (CD3800); 8: *Matteuccia orientalis* (CD3801); 9, 10: *Salvia* sp. (CD3787); 11, 12: negative control (no DNA).

**Figure 7 ijms-25-00844-f007:**
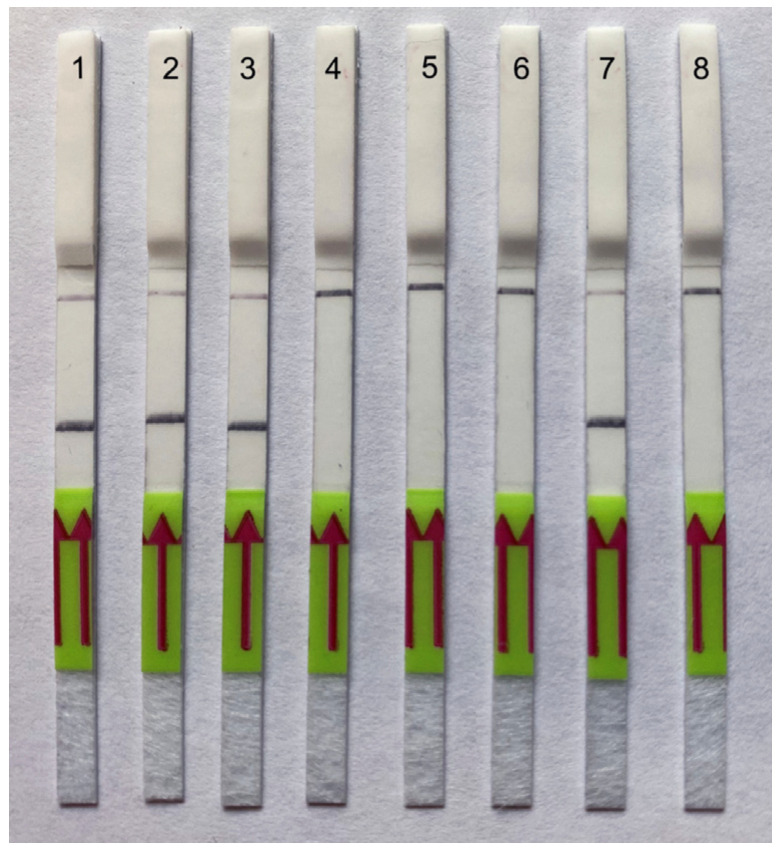
Lateral flow recombinase polymerase amplification (LF-RPA) assay with examples of lateral flow strips. Testing of field samples. Strips: 1–3: nematodes extracted from strawberry plants and *A. fragariae*-infected fern leaves; 4–6: nematodes extracted from strawberry plants; 7: DNA of *A. fragariae* (CD4001); 8: negative control (no DNA).

**Figure 8 ijms-25-00844-f008:**
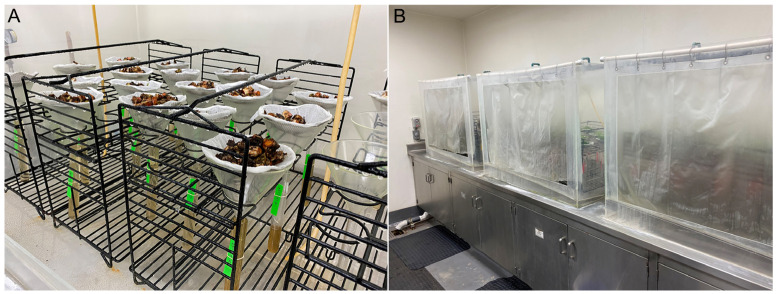
Baermann funnel technique in the mist system for nematode extraction. (**A**) Glass funnels and tubes with strawberry samples. (**B**) Mist extraction system in the Nematology lab, Plant Pest Diagnostic Centre of the California Department of Food and Agriculture.

**Table 1 ijms-25-00844-t001:** Samples of *Aphelenchoides fragariae* and other nematodes tested in the present study.

Species	Locations	Plants	Materials *	Sample Codes	Sources
*Aphelenchoides fragariae*	USA, California, Marin County, Point Reyes National Seashore	*Polystichum munitum*	ND, PD, Ex	CD3774	S.A. Subbotin
*A. fragariae*	USA, California, Mendocino County	*P. munitum*	ND, Ex	CD4001	S.A. Subbotin
*A. fragariae*	USA, Connecticut, Windsor	*Salvia* sp.	ND, Ex	CD3787	J.A. LaMondia
*A. fragariae*	USA, Washington, Clallam County, Storm King Ranger Station	Unidentified plant	ND; Ex	CD3764	S.A. Subbotin
*A. fragariae*	Russia, Moscow, Main Botanical Garden of the RAS	*Pteris* sp.	ND, PD	CA30, CA38, CA53, CD3796	V.N. Chizhov
*A. fragariae*	USA, North Carolina	*Anemone hupehensis*	ND, PD	CD386, CD3797	J.L. McCuiston
*A. fragariae*	USA, North Carolina	*Lantana camara*	ND	CD388	J.L. McCuiston
*A. fragariae*	USA, California	*Oryza sativa*	ND, Ext	CD754	S.A. Subbotin
*A. fragariae*	Germany	*Primula denticulata*	PD	CD3804	D. Sturhan
*A. fragariae*	Germany	*Lagarosiphon cordofanus*	PD	CD3806	D. Sturhan
*A. fragariae*	New Zealand	*Ptisana salicina*	PD	CD3808	D. Sturhan
*A. fragariae*	New Zealand	*Polystichum* sp.	PD	CD3809	D. Sturhan
*A. fragariae*	USA, North Carolina	*Polypodium californica*	PD	CD3794	J.L. McCuiston
*A. fragariae*	USA, North Carolina	*Lantana* sp.	PD	CD3795	J.L. McCuiston
*A. fragariae*	USA, North Carolina	*Anemone* × *hybrida*	PD	CD3799	J.L. McCuiston
*A. fragariae*	USA, North Carolina	*Heuchera* sp.	PD	CD3798	J.L. McCuiston
*A. fragariae*	Germany, Münster	*Mimulus moschatus*	PD	CD3800	D. Sturhan
*A. fragariae*	Germany	*Matteuccia orientalis*	PD	CD3801	D. Sturhan
*A. fragariae*	Germany, München	*Osmunda regalis*	PD	CD3810	D. Sturhan
*A. fragariae*	New Zealand	*Todea barbara*	PD	CD3811	D. Sturhan
*A. fragariae*	Germany, Münster	*Penstemon campanulatus*	PD	CD3934	D. Sturhan
*A. fragariae*	Germany, Münster	*Lithospermum arvense*	PD	CD3944	D. Sturhan
*A. fragariae*	New Zealand	*Blechnum* sp.	PD	CD3950	D. Sturhan
*A. fragariae*	Germany, Münster	*Solidago glomerata*	PD	CD3946	D. Sturhan
*A. fragariae*	Germany, Münster	*Tellima grandiflora*	PD	CD3947	D. Sturhan
*A. fragariae*	Germany, Münster	*Bergenia* sp.	PD	CD3948	D. Sturhan
*A. fragariae*	Germany, Münster	*Ligularia* sp.	PD	CD3949	D. Sturhan
*A. besseyi*	USA, Florida	*Fragaria × ananassa*	ND	CD2415	C. Oliveira
*A. oryzae*	USA, Louisiana, Morehouse Parish, Mer Rouge	*Oryza sativa*	ND	CD2471	C. Overstreet
*A. oryzae*	Russia, Krasnodar	*O. sativa*	ND	CD3790	V.N. Chizhov
*A. pseudobesseyi*	USA, Florida, Sumter County, Sumterville	*Dryopteris erythrosora*	ND	CD2704	W. Crow
*A. pseudobesseyi*	USA, North Carolina, Jackson County, Cullowhee	Soil sample	ND	CD3097	C. Oliveira
*A. pseudobesseyi*	USA, Florida, Alachua County, Gainesville	*Echinacea* sp.	ND	CD2491, CD3702	W. Crow
*A. ritzemabosi*	USA, California, Mendocino County	*Helleborus* sp.	ND	CD1366	S.A. Subbotin
*A. smolae*	USA, Oregon, Bonanza	*Fragaria × ananassa*	ND	CD3775	S.A. Subbotin
*Aphelenchoides* sp.	USA, California, San Diego County	Grasses	ND	CD1300	S.A. Subbotin
*Aphelenchus* sp.	USA, California, Tehama County	*Fragaria × ananassa*	ND	CD3788b	S.A. Subbotin
*Cryptaphelenchus* sp.	Tomsk region, Malinovka	*Abies sibirica*	ND	CD3657	A. Ryss
*Bursaphelenchus* *cocophilus*	Mexico, Guerrero state	*Cocos nucifera*	ND	CD3548, CD3572	I. Cide del Prado Vera
*B. fraudulentus*	Russia, Moscow, Main Botanical Garden of the RAS	*Quercus robur*	ND	CD2935	A. Ryss
*B. mucronatus*	Russia, Buryatia	*Abies sibirica*	ND	CD3642	A. Ryss
*Laimaphelenchus hyrcanus*	Russia, Saint Petersburg	*Q. robur*	ND	CD3646	A. Ryss
Unidentified anguinid nematode	USA, Washington, Clallam County, Storm King Ranger Station	*P. munitum*	ND, Ex	CD3762	S.A. Subbotin

* ND—DNA extracted from nematodes; PD—DNA extracted from infected plants; Ex—crude nematode extract.

**Table 2 ijms-25-00844-t002:** RPA primers and probe for amplification of DNA of *Aphelenchoides fragariae*.

Primer or Probe	Sequence (5′–3′)
AfragF2-ITS	CTT GTT TGA GAT CTT CTA GAC TA
AfragR1-ITS	GCG CCA CATC GGG TCA TTA TTT
AfragR1-ITS-biotin	[Biotin] GCG CCA CATC GGG TCA TTA TTT
Probe-Afrag-ITS-RPA-nfo	[FAM] TG AGT AGT TGT CTA GTT CGT GAC TAC TAA GAC TTT [THF] ATT GGT AGA GTC GCT CTA T [C3-spacer] *

* FAM—fluorophore, THF—tetrahydrofuran, C3—spacer block.

## Data Availability

The data are contained within the article.

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
