# Peer review of "Rapid Detection of the Strawberry Foliar Nematode *Aphelenchoides fragariae* Using Recombinase Polymerase Amplification Assay with Lateral Flow Dipsticks"

_ijms, 2024, doi:10.3390/ijms25020844_

Round 1
Reviewer 1 Report
Comments and Suggestions for Authors
The current study deals with a novel quick and sensitive assay to detect A. fragariae infection in strawberries. The manuscript lacks comparisons and reasoning of how the current approach is better than previous detection methods.
1. For each of the plant samples containing different nematode numbers, a parallel study showing PCR based amplification of the TS sequence would help to correlate and validate the results in Fig 5 and Fig 6.
2. There are already reports of DNA based techniques for the detection of A. fragariae in infected plant samples. A comparison table showing the difference between the current and previous kits would be useful.
3. Appropriate reference must be cited for disease incidence data on A. fragariae, especially in the introduction.
4. I do not see a difference in the sensitivity assay between samples having one nematode and 20 nematodes. Is it due to the saturation of the signal?
5. The data showing RPA assays from infected plant samples, including herbarium specimens is missing.
Comments on the Quality of English LanguageThere is scope for minor improvements in the use of the English language.
Author Response
Thanks for your comments and suggestions
- For each of the plant samples containing different nematode numbers, a parallel study showing PCR based amplification of the TS sequence would help to correlate and validate the results in Fig 5 and Fig 6.
Reply: I included additional sentences about our previously developed PCR assay (McCuiston, J.L.; Hudson, L.C.; Subbotin, S.A.; Davis, E.L.; Warfield, C.Y. Conventional and PCR detection of Aphelenchoides fragariae in diverse ornamental host plant species. Journal of Nematology 2007, 39, 340-350) in Discussion section (lines 205-209). The goal of this study was development of LF-RPA assay for detection of A. fragariae only (line 86), but not make a comparison of LF-RPA assay with PCR assay results. The latter approach requires different methodology and should be considered as different project.
- There are already reports of DNA based techniques for the detection of A. fragariae in infected plant samples. A comparison table showing the difference between the current and previous kits would be useful.
Reply: In our previous publication (McCuiston, J.L. et l., 2007) we provided DNA extraction protocols, in the present study, I used very simple protocol (described in Methods) for nematode crude extraction, it is not DNA extraction protocol.
- Appropriate reference must be cited for disease incidence data on A. fragariae, especially in the introduction.
Reply: Several additional references are included in Introduction
- I do not see a difference in the sensitivity assay between samples having one nematode and 20 nematodes. Is it due to the saturation of the signal?
Reply: LF-RPA assay as well as PCR assay are used for detection, not quantification.
- The data showing RPA assays from infected plant samples, including herbarium specimens is missing.
Reply: These results are given in Figure 6B.
Reviewer 2 Report
Comments and Suggestions for Authors
Rapid Detection of the Strawberry Foliar Nematode, Aphelenchoides fragariae using Recombinase Polymerase Amplification Assay with Lateral Flow Dipsticks
Sergei A Subbotin
Comments to the Authors
This manuscript describes the method development of rapid detection of an important nematode species, Aphelenchoides fragariae in strawberry based on recombinase polymerase amplification and with lateral flow dipsticks. The methods have been described in detail in a manner that can be easily followed. The experimental steps have been conducted appropriately with the relevant controls. The manuscript has very good organization and flow. However, several minor errors were found in the text. The manuscript needs to be thoroughly checked for spelling errors.
Main text:
Introduction:
Line 25: Has a citation that did not get numbered.
Line 58: exanimationà examination
Lines 57-62 Consider adding a citation.
Results:
Lines 190-192: Please add the corresponding figure numbers.
Please report false negative and positive rates of the assay.
Figures:
Figures 1, 5, and 6: Italicize scientific names.
Discussion:
Throughout the text: Please carefully go through the text and Italicize scientific names.
Line 194: Remove additional space between ‘a’ and ‘simple’.
Line 207: Add a space between ‘species’ and ‘that’.
Materials and Methods:
Line 253: Jjuvenileà Juvenile
Line 265: The author states that ‘These species were also identified by molecular methods’. If these methods are referring to the previous methods, either state the methods or add citation.
Comments on the Quality of English Language
The manuscript needs to be thoroughly checked for spelling errors. Please carefully go through the entire text and Italicize scientific names.
Author Response
Thanks for your comments and suggestions!
Main text:
Introduction:
Line 25: Has a citation that did not get numbered.
Reply: corrected.
Line 58: exanimationà examination
Reply: corrected.
Lines 57-62 Consider adding a citation.
Reply: References are added.
Results:
Lines 190-192: Please add the corresponding figure numbers.
Please report false negative and positive rates of the assay.
Reply: Corrected (line 201, 202), New Figure 7 is also included.
Figures:
Figures 1, 5, and 6: Italicize scientific names.
Reply: corrected.
Discussion:
Throughout the text: Please carefully go through the text and Italicize scientific names.
Reply: corrected.
Line 194: Remove additional space between ‘a’ and ‘simple’.
Reply: corrected.
Line 207: Add a space between ‘species’ and ‘that’.
Reply: corrected.
Materials and Methods:
Line 253: Jjuvenileà Juvenile
Reply: corrected.
Line 265: The author states that ‘These species were also identified by molecular methods’. If these methods are referring to the previous methods, either state the methods or add citation.
Reply: References are added.
Comments on the Quality of English Language
The manuscript needs to be thoroughly checked for spelling errors. Please carefully go through the entire text and Italicize scientific names.
Reply: corrected. [Original version of the manuscript contained Italicized scientific names, but then they were converted in a normal font after uploading on the journal server.]
Round 2
Reviewer 1 Report
Comments and Suggestions for Authors
Overall the manuscript shows an easy, quick way of detecting Aphelenchoides fragariae in infected plant samples. However, like my previous comment, I would advise including a comparative approach showing the sensitivity of the kit in comparison to PCR-based detection as well as a crude extraction method involving .03 to 1 nematode per reaction.
In Figure 7, wherein field samples were used for detection, it is strange why in lanes 1-3 in pure Aphelenchoides fragariae infected samples we see a positive band, however in the 4-6 lane there is none at all. The authors have mentioned that this is due to non-specific nematodes present, but Aphelenchoides could be present in a mixed population also. What do lanes 1-3 nematodes from strawberry samples indicate? Does it mean that the nematodes have been isolated from the strawberry plants before their detection or was it done in crude extracts only? A more detailed explanation of the result and figure legend would be useful in this regard.
Author Response
Thanks for your comments:
- The goal of our research is clearly indicated in Introduction and does not include any comparative study with any other methods. As we already underlined, comparative study requires completely different methodology, including long process of PCR optimization, which is used for comparison (testing different polymerases, different PCR termoprofiles, different reagent concentrations ect). Goal of our study is developed RPA assay only. Advantages of RPA over PCR are evident, and we included this sentence in the revised version (line 205): “The LF-RPA assay has some important advantages over our previously developed PCR method of fragariae detection [21], the first being that it uses crude nematode extract for the analysis instead of DNA extracts, which are required for PCR assays. The second advantage is that results are available up to 25 min for RPA vs more than 3.0 h for PCR assay including DNA extraction, PCR and electrophoresis.”
- As for Figure 7. This experiment is described in Methods in details:
Line 278: “For assay validation with field samples, twelve healthy strawberry plants were provided to the CDFA Nematology lab by California growers. Approximately 20 g of plant tissues from each strawberry sample were place in Baermann funnels in mist chambers. One gram of infected fern leaves was added to the funnels of half of these samples (Figure 8). The water gradually filled the collection tubes and overflowed slowly enough that nematodes remained in the bottom of the tube. All samples were incubated under the mist for 2 days and then each collection tube was carefully removed from its funnel without disturbing the contents. A large pipette was used to draw off the water carefully from each tube to avoid stirring up nematodes that had settled to the bottom, as described by Ayoub [44]. Extracts from all samples were visually inspected under a binocular microscope to reveal the presence of moving Aphelenchoides specimens and non-target nematodes. The samples were used to prepare crude nematode extracts.”
Line 317: “For validation of the assay with field samples, live nematodes in 300 mL of water obtained from each sample were added into individual tubes and homogenized with an ultimate laboratory homogenizer, Mini Bead Beater 1 (BioSpec Products, OK, USA) and one glass bead (5 mm) during 30–60 sec to obtain nematode extract. “
Results:
Line 190: “2.4.4. Testing of Field Samples. Detection of the strawberry foliar nematode was confirmed in all strawberry samples with leaves and stolons in which fern leaves infected by A. fragariae were added. All extracts from strawberry samples using Baermann funnels contained more than 30 specimens of non-target nematodes (cephalobids, rhabditids and tylenchids) and strawberry samples with target nematodes contained more than 5 moving A. fragariae specimens, which were visible under a binocular microscope. The samples were used to prepare crude nematode extracts. All samples containing A. fragariae showed strong positive test lines on the LF strips in RPA assays, whereas extracts from strawberry samples containing only non-target nematodes gave only a control line on the LF strips (Figure 7). No false negative or false positive reactions were observed in this experiment.”
Line 179: “Figure 7. Lateral flow recombinase polymerase amplification (LF-RPA) assay with examples of lateral flow strips. Testing of field samples. Strips: 1-3: nematodes extracted from strawberry plants and A. fragariae infected fern leaves, 4-6: nematodes extracted from strawberry plants, 7: DNA of A. fragariae (CD4001) 8: negative control (no DNA).”
Thus: Strips: 1-3: non-target nematodes extracted from strawberry + A. fragariae – Reaction - positive;4-6: no-target nematodes extracted from strawberry plants without A. fragariae - Reaction - negative.
Round 3
Reviewer 1 Report
Comments and Suggestions for Authors
I have no further comments.